# Development of 3D-Formed Textile-Based Electrodes with Flexible Interconnect Ribbon

**DOI:** 10.3390/s25020414

**Published:** 2025-01-12

**Authors:** Paula Veske-Lepp, Glenn Van Steenkiste, Svea Thienpondt, Joris Cools, Herbert De Pauw, Frederick Bossuyt

**Affiliations:** 1Centre for Microsystems Technology (CMST), IMEC and Ghent University, 9050 Zwijnaarde-Gent, Belgium; paula.veske@gmail.com (P.V.-L.); herbert.depauw@ugent.be (H.D.P.); 2Equine CardioTeam, Department of Internal Medicine, Reproduction and Population Medicine, Faculty of Veterinary Medicine, Ghent University, 9820 Merelbeke, Belgium; glenn.vansteenkiste@ugent.be; 3Fashion and Textiles Innovation Lab (FTILab+), HOGENT University of Applied Sciences and Arts, 9051 Ghent, Belgium; svea.thienpondt@hogent.be (S.T.); joris.cools@hogent.be (J.C.)

**Keywords:** e-textiles, electrodes, smart textiles, conductive textiles, electronics, IoT

## Abstract

The integration of electronics into textiles has gained considerable attention in recent years, due to the development and high demand of wearable and flexible electronics. One of the promising fields is healthcare, which often involves the utilization of textile-based electrodes. These electrodes often offer advantages such as conformability, breathability, and comfort. This article presents the development of 3D-formed textile-based electrodes together with a narrow fabric-based interconnect system. This study showcases the methods and materials for the fabrication of the textile-based electrodes and the interconnect system, including a durability assessment, by performing standardized washing (ISO 6330-2012) and user tests. The results demonstrated that the developed 3D-formed textile-based electrodes and stretchable interconnect system are durable and effective for wearable applications, maintaining performance under extensive washing.

## 1. Introduction

The convergence of textiles and electronics has paved the way for innovative solutions in wearable technology. Traditional rigid electronic components are being replaced by flexible and conformable alternatives to enhance user comfort and functionality. The integration of textiles and electronics has garnered significant attention due to the inherent flexibility, breathability, and lightweight nature of textile substrates, which are inherently suitable for prolonged wear. Among these alternatives, textile-based electrodes emerged as a promising platform due to their soft and conformable properties. Electrodes can be used in various fields, including continuous care [1], rehabilitation [2], sports [3], work environments [4,5], prosthetics control [6], and gaming [7]. Moreover, the utilization of 3D-forming techniques through thermoforming enabled the creation of different structures that conform precisely to body surfaces, ensuring optimal contact and signal acquisition, even in dynamic and irregular environments [8,9,10].

Previously, many 3D electrodes were fabricated through knitting methods [11,12,13], embroidery [14] or by using soft, inherently 3D materials, such as foam materials, under conductive textiles [15,16,17]. The design of dry textile electrodes still faces the same challenges, such as compensating for signal interference [18]. Interference and noise are often the first problems encountered; therefore, they are also mostly addressed. For example, Puurtinen, et al. [19] reported that noise levels increase with decreasing electrode sizes. Moreover, Maji and Burke [20] showed that un-gelled fabric-based electrodes do not appear to be intrinsically noisier than gelled electrodes when used with amplifiers with low input noise currents and, thus, encourage the research on textile-based electrodes further. In order to reduce the impedance during use and also maintain comfort, it is imperative to find the electrodes’ optimized size, shape, use conditions, and material stack.

Washability is another challenge for textile-based electrodes. High resistance to washing can increase their usability together with supporting higher durability and reliability, as well as make them more environmentally friendly. Arquilla et al. [21] washed silver-based textile electrodes eight times in a non-standardized method with low resistance changes. The washability of textile electrodes could increase sustainability (prolonged use of electrodes) and user convenience during reuse.

The incorporation of flexible interconnects provides seamless connectivity between individual electrodes, enhancing overall system reliability and comfort [22,23]. Interconnects as ribbons were used many times, often with snap fasteners and in a flexible and stretchable or non-stretchable way [24,25]. However, ribbon interconnects can pose several challenges, such as susceptibility to structural stability fatigue and breakage over time. Often, there is a loss of connection due to the snap button fatigue or weak/unstable connections between some other connector types. Ensuring durability and maintaining reliable performance in different environments over time remain significant obstacles.

This study aimed to show the fabrication of flexible, stretchable interconnects and soft, adaptable 3D-formed electrodes (used and tested currently only on animals) that can be sewn or laminated into a range of products. This work presents the fabrication of electrodes and the interconnect system. This research demonstrated their durability against physical degradation and washing.

## 2. Materials and Methods

### 2.1. Three-Dimensionally Formed Electrodes

Table 1 characterizes the materials used to fabricate the electrodes. The conductive layer was from conductive fabric (Shieldex^®^, Bremen, Germany) that was laminated with 250 µm thermoplastic polyurethane (TPU) (Prochimir, Sèvremont, France). The laminated conductive fabric was then thermoformed using a negative mold, whereas the fabric side touched the mold. The backside of the thermoformed sample (TPU-side) was then laminated with knitted fabric (Eurojersey, Caronno Pertusella, VA, Italy) that was also laminated with the same parameters beforehand with 100 µm TPU (Bemis, Neenah, WI, USA). The conductive fabric was placed in the same polytetrafluoroethylene (PTFE) mold before the final lamination to keep the thermoformed shape. See the process flow on Figure 1 with descriptive photos.

Lamination was carried out on a Hotronix^®^ Air Fusion IQ^®^ heat press (STAHLS’ Europe, Dillingen, Germany), and rigid FR4 boards were placed on the lower plate to avoid air bubbles during the lamination. The lamination parameters were 170 °C at 4 bar pressure for 40 s. Thermoforming was carried out using Formech 300XQ (Formech Europe GmbH, Leimersheim, Germany) at ~200 °C and heated for 40 s.

### 2.2. Interconnect

Table 2 describes the materials used to fabricate the flexible, stretchable interconnect. Conductive yarn (Bekaert, Zwevegem, Belgium) was weaved onto a stretchable narrow fabric (Liebaert, Deinze, Belgium) in a horse-shoe meander shape. Every 5 cm, the meander shape is extended to create a longer yarn opening/detachment from the ribbon (Figure 2). This was needed to so that the ribbon can be produced continuously but has the flexibility to be cut at a certain length and be connected to the snap button with a soft connector. The endings of the ribbons included a stack of TPU, conductive fabric, and low melting point solder paste (Figure 3). The interconnect endings were finished in two steps: (1) the bottom laminated fabric layer, laminated conductive fabric, and ribbon with conductive yarn were combined by laminating them together, with a TPU layer featuring a hole added on top; (2) low-melting-point solder paste was applied to the exposed yarn (through the hole in the TPU layer), followed by placing the top laminated conductive fabric and laminated fabric layers, completing the final lamination.

### 2.3. Durability Testing

The testing of the electrodes and the interconnect system included washing tests described in Table 3. There were 4 samples (each 25 cm long) in each sample set, and the interconnect narrow fabric pieces were sewn onto a knitted fabric. The samples were put into a meshed washing bag prior to the tests. The samples’ functionality was checked every washing cycles until 5 cycles were completed and then after every 5 washing cycles. The functionality check was completed with a multimeter. The electrodes were measured from the right side of the electrodes on the edge of the fabric to the back side of the electrode from the snap fastener. Electrodes measurements were reported as measured (in Ohms). The interconnect systems were measured from one ending from the snap fastener to another ending (Figure 4). Interconnect system measurements were calculated to be Ohms per meter.

The samples were washed until failure or to 50 cycles, but the goal was to reach minimum 25 washing cycles without losing any functionality (resistance would stay under 10 ohm per meter per measurement for both electrodes and interconnect samples). The washing tests were performed according to ISO 6330-2012 standard (Geneva, Switzerland) at 60 °C. The high temperature was requested by the end-user of specific application (veterinary field). The washing tests always included type 3 detergent from ISO 6330-2012 standard (Geneva, Switzerland). The mechanical durability or structural integrity of the samples was tested through a washing process as the process included heavy spinning several times, as seen in Table 3.

### 2.4. Testing of the Electrodes

This study was approved by the ethics committee of the Faculty of Veterinary Medicine and Bioscience Engineering (EC2020/65). For this study, electrode performance was assessed on three horses during different seasons to evaluate the effects of varying coat lengths. Electrodes were placed in a modified base apex configuration, and a commercial veterinary ECG recorder was used (Televet 100, Engel Engineering GmbH, Schwertberg, Austria) [26]. Prior to application, each electrode was wetted by brief submersion in tap water and then secured using a body bandage (Norton Compression, Jamesburg, NJ, USA). Short-term tests were conducted under two conditions: the horses were left free in the stable for 10 min and also manually walked up and down a corridor.

The tests took place in late summer (September), when the horses had a shorter coat, and in winter (December), when they had a longer coat. Additionally, a long-term test was performed in the summer, where electrodes were applied and remained in place for up to 24 h, or until signal loss occurred. Impedance measurements were conducted for assessing the conductivity between the skin and electrodes; the electrodes were positioned 10 cm apart (conductive edge to edge), with measurements taken at 10 kHz (in order to avoid nerve stimulation) using an LCR meter 10 min post-application to ensure stabilization of the skin- electrode interface (Figure 5). These results were compared with AgCl electrodes (Skintact FS-50, Leonhard Lang GmbH, Innsbruck, Austria) to establish performance benchmarks.

## 3. Results

### 3.1. Electrodes

The 3D-formed textile-based electrodes were successfully fabricated using the specified lamination and thermoforming process. The process resulted in electrodes that maintained their conformable shape and structural integrity. Incorporating conductive fabric (Shieldex^®^) laminated with TPU and subsequently integrated with a secondary knitted fabric (Eurojersey) demonstrated effective layering without delamination.

The durability of both the electrodes and interconnect system was evaluated through extensive washing tests following the ISO 6330-2012 standard at 60 °C using type 3 detergent. Four electrode samples were tested for washing durability. The samples were subjected to washing cycles to simulate real-world usage conditions, with functionality checks performed after every cycle until five cycles, and subsequently after every five cycles. All electrode samples withstood a minimum of 50 washing cycles without any loss of functionality. The resistance remained below 10 Ω throughout the testing period, meeting the durability criteria set for the study. After 10–15 washing cycles, a slight increase in resistance was observed (average resistance reached 2 ± 1 Ω), but this remained within acceptable limits for practical applications. As resistance measurements were conducted, the snap fastener movement on the sample and electrode material stretching caused the resistance stability to decline. However, after 30 washing cycles, some delamination occurred around the edges of the electrodes (Figure 6).

The impedance of the proposed electrodes was 54.1 kΩ, 32.27 kΩ, and 8.02 kΩ (Figure 7), compared to the AgCl electrodes impedance of 51.1 kΩ, 11.9 kΩ, and 15.9 kΩ, respectively. Short-term recordings showed subjectively usable ECG compared to AgCl on all horses on both winter and summer coat during free movement in the stable (Figure 8). However, movement artifacts made the ECG uninterpretable during walking. Long-term tests enabled ECG recording of 7 h 43 min, 15 h 54 min, and 24 h before the ECG signal faded, and no further interpretation was possible, possibly because the electrodes/fur became dry, and the impedance became too high for further signal transfer. The horses with a subjectively shorter coat had longer recording times. ECG tests showed no significant changes in basic electrode characteristics; resistance from snap fastener to conductive layer remained stable and no visible damage was observed.

### 3.2. Interconnect System

The flexible and stretchable interconnect system, fabricated using conductive yarn (Bekaert) on a stretchable narrow fabric (Liebaert) in a horse-shoe meander configuration, demonstrated robust electrical connectivity and sample structural durability. The design allowed for both flexibility and stretchability, critical for wearable applications.

The initial electrical resistance of the interconnects ribbon with processed endings was recorded at an average of 8.0 ± 0.5 Ω per 1 m segment. The meander design provided a stretchable and uniform conductive path as indicated by consistent resistance values across different interconnect samples. Snap fasteners were also easily added and did not show any significant electrical losses during attachment or detachment. For testing, the dry electrodes were attached and detached from the interconnect system endings 50 times. This did not degrade the connection of the snap fasteners on either of the dry electrodes or the interconnect system endings.

After the minimum of 25 washing cycles, the shape or electrical resistance measurements of the interconnect samples stayed between 8 and 20 Ω/m, but with the exception of one sample (Figure 9). The resistance remained below 20 Ω/m throughout the testing period for three samples, meeting the durability criteria set for the study. After 40 washing cycles, instability in measuring resistance was observed in another sample (resistance was between 10 and 100 Ω). In general, two samples from four remained stable until 50 cycles had passed, one sample showed degradation around the snap fastener (snap fastener itself was attached) after 20 cycles, and one sample after 40 cycles.

Beyond 15 washing cycles, all the interconnect samples began to exhibit minor signs of degradation, such as slight fraying from the edges and starting to show delamination (Figure 10). These brought out high and more sudden resistance changes (around 2–2.5 times). It was observed that the delamination resulted in weaker connections between conductive adhesives, conductive yarns, and fabrics. However, these issues did not result in immediate functional failure, as the samples continued to pass the functionality checks with one sample exception. As previously mentioned, another failure started to occur around 40 cycles. It was clear that the electrical connection between the yarn/ribbon ending and the snap fastener was broken due to the mechanical wear that was caused in the washing machine drum during spinning and other washing process steps as the yarn itself was still conducting electricity fine.

## 4. Discussion

The results indicated that the 3D-formed textile-based electrodes and the flexible, stretchable interconnect system developed in this study are effective in certain scenarios for wearable electronic applications. The interconnect system shows acceptable durability to the washing process (samples could be washed over 20 times) with room for improvement in the sample set stability. The combined electrode and interconnect system provide a structurally stable and durable solution, suitable for applications requiring frequent washing and flexibility, such as in healthcare monitoring for animal care. However, it is important to note that the wearable system had considerable noise during larger movements, like walking. This issue still needs to be tackled in future work.

The conductive yarn encapsulated between layers of TPU and fabric had lost its connection and/or corroded enough to cause a drastic increase in resistance. This could be from insufficient bonding materials and parameters. To improve interconnect durability, one potential solution is to test alternative materials, such as less slippery ribbon material or different TPU materials for better lamination results. Also, an alternative approach would be to implement a more flexible design for the interconnects that could help absorb mechanical stresses (forces caused by washing machine drum and spinning steps in the washing process) and reduce the risk of disconnection.

Nevertheless, the bill of materials could be decreased for sustainability and cost purposes. For example, the non-conductive knitted fabric for the electrodes backing and the interconnect endings’ encapsulation can also be laminated with Prochimir TPU to only have one TPU type and to increase stack stability. In general, the interconnect stack consists of several materials which create more waste and should be minimized in the future developments. However, then, the washing reliability needs to be re-tested to make sure there is no delamination of the stack.

The electrodes exhibited good resistance to washing, maintaining electrical functionality after prolonged exposure to mechanical stress and detergent. The minimal increase in resistance up to 50 cycles indicates that the current materials and materials’ stack were effective. However, some instances of delamination and edge degradation were observed, particularly after 30 cycles. The delamination observed may have resulted from insufficient adhesion at the edges, which experience the highest mechanical stress during washing and handling. One potential solution to mitigate these issues is to explore more the use of different types TPU film, similarly to the interconnect system, and explore more bonding process parameters.

It is also important to highlight opportunities for enhancing sustainability by reducing the number of materials and minimizing waste. Currently, the system employs multiple layers, including a non-conductive backing and encapsulation material for the interconnects, which could be streamlined. By unifying the TPU materials used across the electrode and interconnect stack, there could be a reduction in the overall material consumption and, consequently, production costs. The waste of leftover electrode stack after final laser should be optimized. The mold for thermoforming could be optimized further. To produce less waste, the laminated conductive fabric and the final electrode stack could also be optimized before final laser cutting. Alignment and layering process could be improved further to ensure uniform thickness, quality and conductivity throughout the fabric. Additionally, the production process could be automated to reduce labor costs and improve efficiency.

During short-term recordings, the electrodes were able to capture usable ECG data of horses under both winter and summer coat conditions within stables despite the issue of movement artifacts during walking. This suggests a promising initial design yet highlights the need for refining the structural integrity stability and surface adherence to reduce artifact interference. Significantly, long-term tests demonstrated variability in recording durations—ranging from 7 h 43 min to 24 h—implying that the electrodes’ performance may be compromised by drying out, while the AgCl electrodes remain wet under the non-breathable and tightly sealed (adhesive) barrier. Thicker coats may require higher moisture levels to remain sufficiently conductive. The variability in recording duration across different measurement times likely stems from differences in coat thickness, skin moisture levels, and the resulting electrode–skin impedance. This suggests that while the current design performs well under ideal conditions, adjustments to electrode size, contact pressure, or fixation materials may be necessary to ensure consistent performance across varying coat and skin conditions. Future work could also consider environmental factors like humidity and temperature, which may further impact electrode function over time. To mitigate these, we proposed additional testing with moisture-retaining materials with the aim to maintain consistent electrode performance under diverse environmental conditions and coat types.

Notably, most other studies using dry electrodes on horses at rest conduct recordings for only up to 1 h [27,28]. Unlike previous studies that achieved recording durations of up to 1 h with dry electrodes, this design enabled recordings of up to 24 h. However, these results suggest that our dry electrode solution is at least as effective as other published solutions. In another study with dry electrodes on horses [29], the horses were first clipped at the electrode sites. However, this is usually not desirable by owners in the field, and we, therefore, did not do this in our study. Another factor affecting the signal stability of dry electrodes is contact pressure and electrode size, both of which were currently not controlled or varied in the current study [29,30].

Furthermore, (dry) electrodes are particularly sensitive to the relative movement of the skin to the electrode [31]. Better adapted fixation materials, larger electrodes, active filtering methods [27] for movement artifact and/or perhaps higher local pressure could further reduce motion artifacts. Manually varying the pressure on the electrodes did indeed influence the impedance of the electrodes. However, the pressure during the impedance measurements was more or less constant due to the use of the same tension girth, so pressure is not likely to be the only factor for the large differences in impedance and skin type will also play a large role. For example, in humans, it is known that skin types with a higher humidity have a lower impedance [32]. Indeed, the impedance of the electrodes in two of the horses was within the range of what another study has measured using dry electrodes on human skin [33], suggesting that our setup operates within known parameters when conditions are favorable. However, in the third horse, the impedance was much higher. This discrepancy indicates that there may be substantial individual variation in skin properties among horses, analogous to the variations observed in humans with different skin types. Identifying and understanding the factors that contribute to these differences could be crucial in further optimizing electrode design and application techniques for more consistent results across diverse subjects.

## 5. Conclusions

In conclusion, 3D-molded textile-based electrodes represent a promising platform for the integration of electronics into textiles. Their unique properties, including flexibility, breathability, and washability, make them well-suited for long- and short-term wear. This study demonstrated the successful fabrication of flexible, stretchable interconnects and soft, adaptable 3D-molded electrodes that are suitable for sewing or lamination into various products. This study demonstrated the development of both the electrodes and the interconnect system, emphasizing their robust performance and durability. Notably, the electrodes exhibited structural integrity and stable electrical performance, maintaining functionality after up to 50 washing cycles, with only minor delamination and slight increases in resistance observed after 30 cycles. This result offered a promising solution for washable and reusable textile electronics in healthcare monitoring. However, issues such as movement artifacts during walking and variability in long-term ECG recording durations were seen, with performance influenced by coat length, skin properties, and electrode–skin contact. These findings emphasized the need for further refinements in fixation materials, and electrode design and final material choice to improve signal reliability under dynamic conditions. Additionally, optimizing the interconnect system, particularly around the snap fasteners and TPU lamination, could further enhance durability and reduce the risk of disconnections during prolonged use and cleaning processes.

Also, this study identified opportunities for improving sustainability and cost-efficiency. It is possible to reduce manufacturing costs and environmental impact in future iterations by reducing material usage, reducing waste, and automating production processes. As part of the future work, the variety of materials will be limited in the bill of materials to create a more homogeneous design. Furthermore, a greater emphasis will be placed on reducing waste material by optimizing the cut outs of the original material and the final laser cutting process.

## 6. Patents

Two patent applications were submitted based on this work—EP24194867.8 and EP24180869.0.

## Figures and Tables

**Figure 1 sensors-25-00414-f001:**
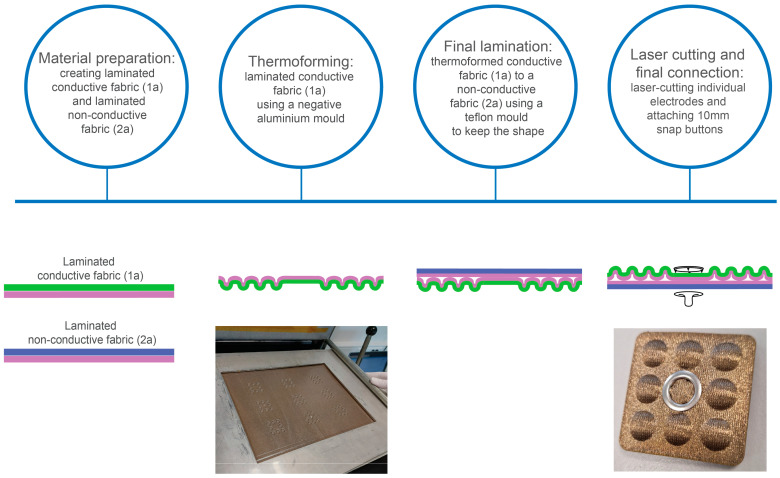
The process flow of electrodes preparation in 4 steps: material preparation; thermoforming; final lamination; and laser cutting with final connection. Cross-section of material stacks and descriptive photos were added to the steps. TPU layer is shown as purple, conductive fabric as green, non-conductive fabric as blue.

**Figure 2 sensors-25-00414-f002:**
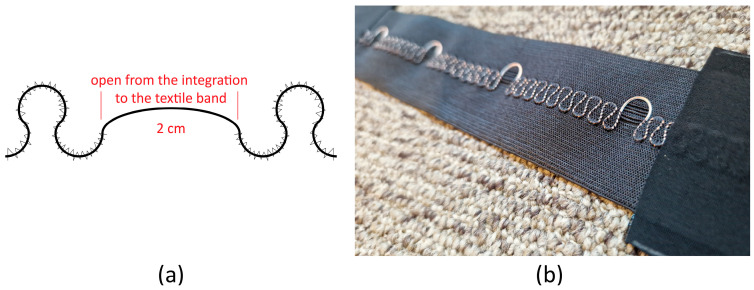
Interconnect ribbon with conductive yarn that was attached onto ribbon in a horse-shoe shape, each sample was 25 cm long. The horse-shoe shape was extended every 5 cm. (**a**) Technical drawing of the yarn attachment to the ribbon. (**b**) Photo of the ribbon with yarn.

**Figure 3 sensors-25-00414-f003:**
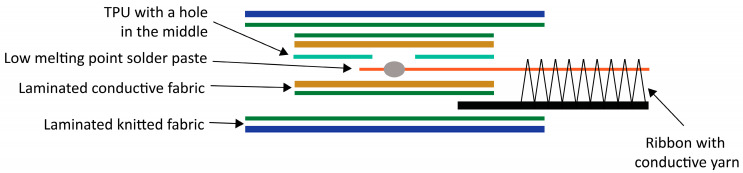
Cross-section drawing of the ribbon ending/connector.

**Figure 4 sensors-25-00414-f004:**
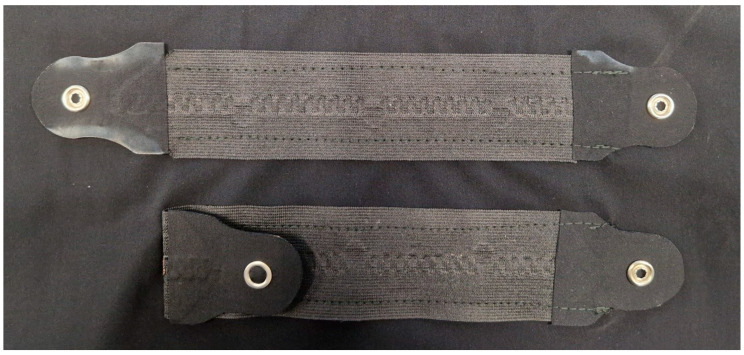
Part of the washing sample for the interconnect system.

**Figure 5 sensors-25-00414-f005:**
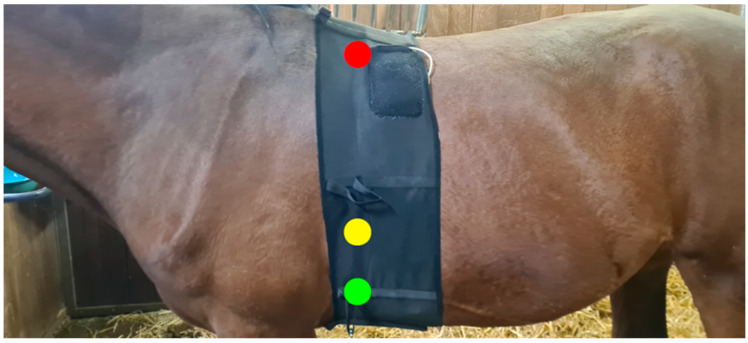
ECG electrode positions on a horse using a modified base-apex configuration. Electrode colors follow the IEC color code: red for the right arm electrode, yellow for the left arm electrode, and green for the left leg electrode. All electrodes are placed underneath the body bandage, with the ECG recorder inside a pouch sewn onto the body bandage.

**Figure 6 sensors-25-00414-f006:**
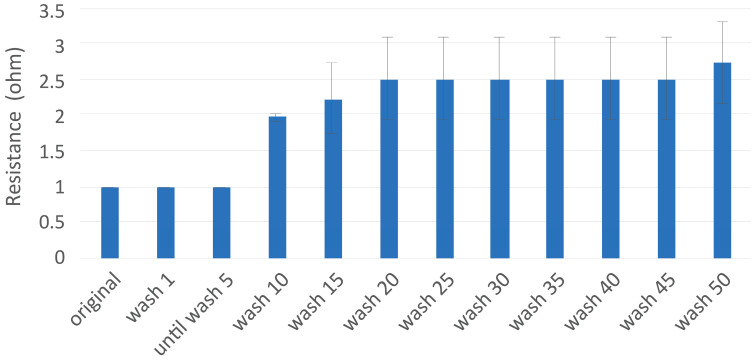
Washing test results of electrodes. The resistance values are taken as the average measurements of all the samples. The resistance was measured from the right side of the electrodes on the edge of the fabric to the back side of the electrode from the snap fastener.

**Figure 7 sensors-25-00414-f007:**
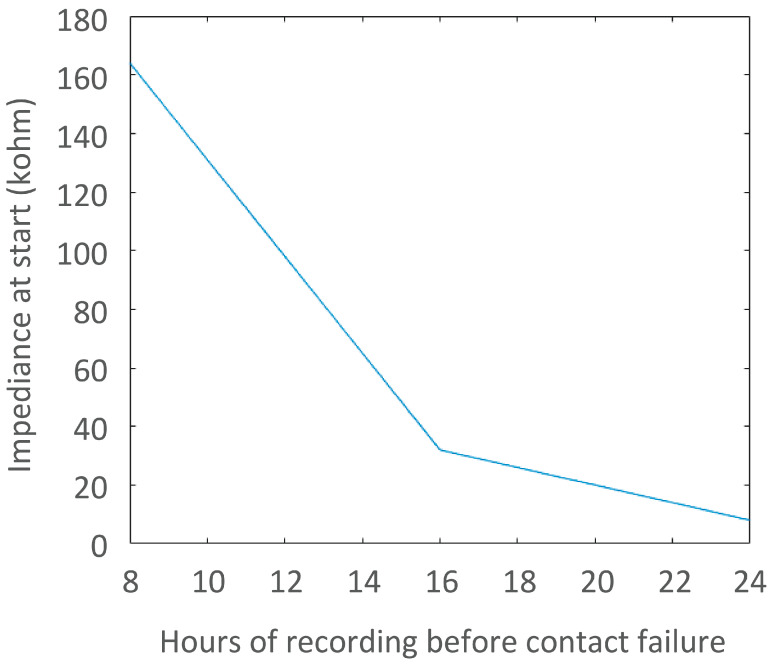
Relationship between hours of successful ECG recording (*x*-axis) and initial impedance (10 kHz) of the textile electrodes measured 10 min after application (*y*-axis). A clear increase in recording time can be seen with a decrease in impedance.

**Figure 8 sensors-25-00414-f008:**
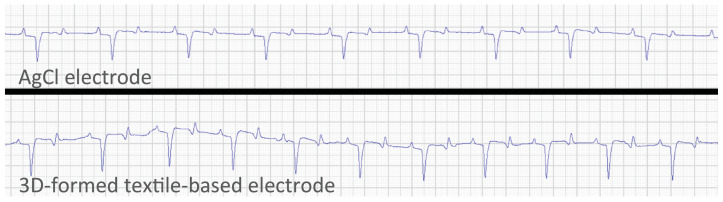
Comparison between a sequential recording with the proposed electrodes and traditional AgCl electrodes recorded at 10 mm/mV and 25 mm/s.

**Figure 9 sensors-25-00414-f009:**
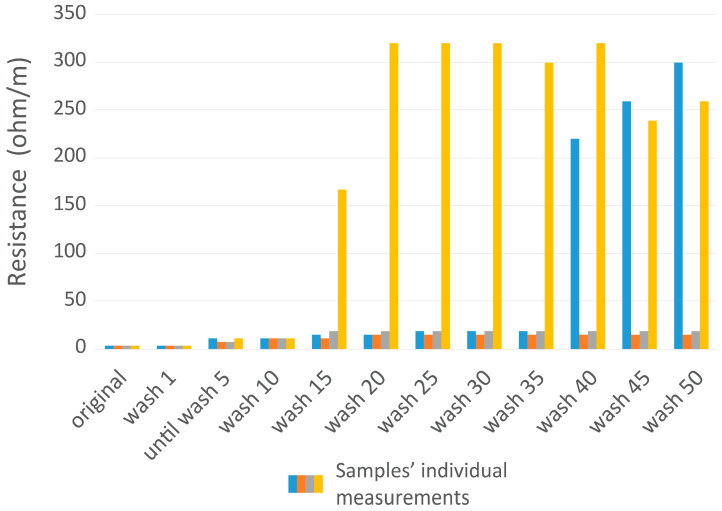
Washing test results for the interconnect system. Resistance values are shown for all samples individually as there were 2 outliers in the sample set.

**Figure 10 sensors-25-00414-f010:**
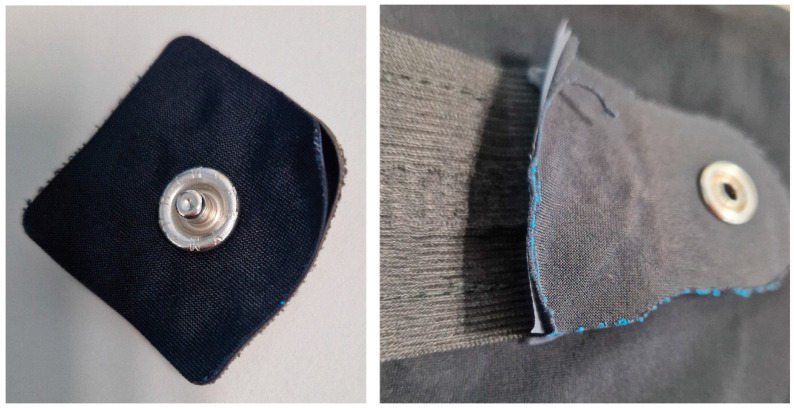
Degradation of lamination on electrode and the interconnect system samples.

**Table 1 sensors-25-00414-t001:** Description of materials used for electrodes. Materials 1a and 2a are composite materials assemble by authors.

Nr	Material	Description	Composition	Additional Properties
1	Knitted conductive fabric	Shieldex^®^ Technik-tex P180 + B	94% polyamide + 6% Dorlastan Metal plated with 99.9% Pure Silver (the metal content of the fabric 26.83 +/−1%)	~205 g/m^2^ Surface resistance < 2 Ω/m^2^ Coating with nitrile rubber
1a	Laminated knitted conductive fabric	Shieldex^®^ Technik-tex P180 + B with Prochimir TC5011	94% polyamide, 6% Dorlastan + TPU	~205 g/m^2^ with polyurethane
2	Knitted fabric	Eurojersey Sensitive Fit	68.0% polyamide 32.0% elastane	213 g/m^2^
2a	Laminated knitted fabric	Eurojersey Sensitive Fit with Bemis 3914	68.0% polyamide 32.0% elastane	~213 g/m^2^ with polyurethane

**Table 2 sensors-25-00414-t002:** Description of materials used for interconnect. Materials 1a and 2a are composite materials assembled by authors.

Nr	Material	Description	Composition	Additional Properties
1	Knitted conductive fabric	Shieldex^®^ Technik-tex P130 + B	78% polyamide, 22% elastane, Metal plated with 99% Pure Silver	~130 g/m^2^ Surface resistance < 2 Ω/m^2^ Coating with nitrile rubber
1a	Laminated knitted conductive fabric	Shieldex^®^ Technik-tex P130 + B with Bemis 3914 TPU	78% polyamide, 22% elastane + TPU	~130 g/m^2^ with polyurethane
2	Knitted fabric	Eurojersey Sensitive Fit	68.0% polyamide 32.0% elastane	213 g/m^2^
2a	Laminated knitted fabric	Eurojersey Sensitive Fit with Bemis 3914	68.0% polyamide 32.0% elastane	~213 g/m^2^ with polyurethane
3	Thermoplastic film 2	Prochimir 5011	Polyurethane	250 µm thickness, adhesive only
4	Stretchable narrow fabric	Knitted, 30 mm wide	100% polyester	Conductive yarn carrier
5	Conductive yarn	Bekiflex^®^ (Bekaert, Zwevegem, Belgium), 14 copper-plated filaments	Steel filaments coated with copper and with general PFA-coating	Interconnect between electrodes and the read-out system
6	Snap fastener	Prym (Stolberg, Germany), 10 mm 2-part snap fastener	100% steel	
7	Low melting point solder paste	Interflux^®^ (Gent, Belgium) DP 5600	Sn42Bi57Ag1	Melting point 139 °C

**Table 3 sensors-25-00414-t003:** Description of washing test process. The washing tests were conducted according to ISO 6330-2012 standard at 60 degrees.

Nr	Washing Phase	Time, min	Temperature, °C	Spin, Ramp (Revolutions per min)	Water Volume, L
1	Main wash, detergent nr 3	20	60	49	15
2	Rinse 1	3	Coldwater	49	19
3	Drain 1	3	Coldwater	49	-
4	Rinse 2	3	Coldwater	49	19
5	Drain 2	3	Coldwater	49	-
6	Rinse 3	3	Coldwater	49	19
7	Drain 3	3	Coldwater	49	-
8	Rinse 4	3	Coldwater	49	19
9	Drain 4	3	Coldwater	49	-
10	Spin	5	Coldwater	1100	-

## Data Availability

The original contributions presented in this study are included in the article. Further inquiries can be directed to the corresponding authors.

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
