# Peer review of "Development of 3D-Formed Textile-Based Electrodes with Flexible Interconnect Ribbon"

_sensors, 2025, doi:10.3390/s25020414_

Round 1

Reviewer 1 Report

Comments and Suggestions for Authors

This authors presents the development of 3D-formed textile-based electrodes together with a narrow fabric-based interconnect system. The topic is interesting and promising, however I would recommend to a possible publication provided with the satisfactory revision of the following concerns:

1. The references of the articles could be improved. There are many research groups, contributing on e-textile research with high quality articles published in high impact journals. The authors might consider to  improve the references to update their literature. 

2. The presentation of the figures in the articles are not consistent. Some of the figures carrying legends in bold letters, some are with very big fonts. Overall, it is not carefully presented in the manuscript.

Comments on the Quality of English Language

N/A

Author Response

This authors presents the development of 3D-formed textile-based electrodes together with a narrow fabric-based interconnect system. The topic is interesting and promising, however I would recommend to a possible publication provided with the satisfactory revision of the following concerns:

  • The references of the articles could be improved. There are many research groups, contributing on e-textile research with high quality articles published in high impact journals. The authors might consider to  improve the references to update their literature. 

References have been updated, nothing below 2010.

  • The presentation of the figures in the articles are not consistent. Some of the figures carrying legends in bold letters, some are with very big fonts. Overall, it is not carefully presented in the manuscript.

Figures were checked and changed to be more consistent. One figure was added for more clarity.

Reviewer 2 Report

Comments and Suggestions for Authors

The authors presented quite interesting results that can be implemented right now. Despite this, the quality of data presentation is very low. The reviewer had a number of comments.

1) The abstract should be rewritten as it does not provide information about the work done and the results obtained.

2) Page 2 line 53: What is it ECG? Authors should provide explanations for all abbreviations mentioned for the first time.

3) To Tables 1 and 2: What means TPU, PTFE, PA, EA, EL, PES, PFA?

4) Although the authors separated Section 3 Results and Section 4 Discussion, the reviewer did not find an explanation for the results obtained in Section 4.

5) Can the authors explain the reason for the increase in sample resistance by 2-2.5 times after washing?

6) The text of line 173-174 does not correspond well to Fig. 6 or, on the contrary, the data in Fig. 6 does not correspond to the presented text. Why does the x-axis in Fig. 6 start at 8 and not at 0? What causes such changes in the impedance values?

7) To Fig. 8: What is the reason for or what causes the increase in resistance after 15 washes?

8) How were the destruction tests carried out? And the mechanical tests? This information is not in section 2!

9) Section 5 is as uninformative as the Abstract!

The above comments do not detract from the high level of the work but will help readers better understand the results obtained.

Author Response

The authors presented quite interesting results that can be implemented right now. Despite this, the quality of data presentation is very low. The reviewer had a number of comments.

  • The abstract should be rewritten as it does not provide information about the work done and the results obtained.

The abstract was edited with more specifics about the work.

  • Page 2 line 53: What is it ECG? Authors should provide explanations for all abbreviations mentioned for the first time.

An explanation has been added, as well for the other abbreviations.

  • To Tables 1 and 2: What means TPU, PTFE, PA, EA, EL, PES, PFA?

Indeed, we apologize for the oversight. Everything is now written in full or the abbreviations are explained in the brackets. For TPU, it is explained when it is first mentioned in the text.

  • Although the authors separated Section 3 Results and Section 4 Discussion, the reviewer did not find an explanation for the results obtained in Section 4.

More explanations were added to Section 4.

  • Can the authors explain the reason for the increase in sample resistance by 2-2.5 times after washing?

We explained further the resistance rise in some samples for the interconnect system (in Section 3, after the Figure 5 and 8).

  • The text of line 173-174 does not correspond well to Fig. 6 or, on the contrary, the data in Fig. 6 does not correspond to the presented text. Why does the x-axis in Fig. 6 start at 8 and not at 0? What causes such changes in the impedance values?
    Thank you for your feedback. There appears to be some confusion regarding Figure 6 (now Figure 7) and the accompanying text. The x-axis in Figure 6 represents the hours of successful ECG recording before contact failure, not starting from zero hours since initial readings are not relevant for this display. The y-axis indicates the initial impedance of the electrodes measured 10 minutes after application. This has been clarified in the figure legend to better align with the data presented in the text and improve clarity for the readers. The changes in impedance values reflect different levels of electrode performance, as lower impedance generally correlates with longer recording times.
  • To Fig. 8: What is the reason for or what causes the increase in resistance after 15 washes?

We explained further the resistance rise in some samples for the interconnect system (in Section 3, before the Figure 6 and after the Figure 9).

  • How were the destruction tests carried out? And the mechanical tests? This information is not in section 2!

All mechanical wear was tested through washing tests as these include water/washing detergent exposure, and heavy spinning in the drum. This was further explained throughout the article, where mechanical was mentioned and at Section 2.3.

  • Section 5 is as uninformative as the Abstract!

Section 5 was improved by more specific information.

The above comments do not detract from the high level of the work but will help readers better understand the results obtained.

Reviewer 3 Report

Comments and Suggestions for Authors

This study reported a textile-based electrode with good electrical properties and structural integrity while maintaining mechanical flexibility through well-interconnected configurations. The potential of the resulting textile electrode system was demonstrated by recording the electrical activity of the horses’ heart rhythms with ECG device under varying test conditions. In particular, the formed textile electrodes exhibited good structural durability after washing, suggesting their potential for applications in everyday life. However, the performance also seems to be highly dependent on the textile itself and the sewing skill rather than on the incorporated electrode. In addition, some issues need to address to clarify the results and support publication in Sensors. The reviewer’s comments on the present study are as follows.  

Please provide schematics or pictures of the testing systems (with ECG recorders) to enhance readers’ understanding.

The authors explained that the aim of the research is to enhance the impedance of textile electrodes through the implementation of specific interconnect systems. However, the results and corresponding discussion are insufficient to clearly address this issue. For example, in Figure 6, the durability of the ECG recorders appears to vary depending on the initial impedance values of the textile electrodes. What is the possible reason for the different start impedance values of each electrode? Please provide the specific solutions to enhance the electrical properties (or impedance) of the textile electrodes in such cases.

The authors need to provide the electrical properties of the conductive epoxy. As mentioned by authors, the electrical conductivity is the most significant factor in ensuring reliable electronic signals. This indicates that the intrinsic electrical conductivities of all components and the contact resistance between them are also crucial determinants of the results.

Long-term ECG recording tests were conducted for 7h 43min, 15h 54min, and 24h until signals were no longer detected. The authors suggested that this possibly due to an increase in impedance at the electrode/fur interface. What were the differences in testing conditions for the electrodes with varying ECG recording durations?

What is the typical standard for excellent mechanical durability? There are no references for comparison with the results.

Specific data demonstrating the durability of textile electrodes during long-term ECG recording, in any form, may be required.

If the formed electrode system can be applied to human skin, the data trends may differ somewhat from those obtained from horses with fur. Is the goal of the present electrode system intended solely for use with animal? If so, it is necessary to clearly specify the requirements for such measurement conditions.

Please check whether the unit ’degrees’ at line 122, should be ‘degree Celsius’.

Author Response

This study reported a textile-based electrode with good electrical properties and structural integrity while maintaining mechanical flexibility through well-interconnected configurations. The potential of the resulting textile electrode system was demonstrated by recording the electrical activity of the horses’ heart rhythms with ECG device under varying test conditions. In particular, the formed textile electrodes exhibited good structural durability after washing, suggesting their potential for applications in everyday life. However, the performance also seems to be highly dependent on the textile itself and the sewing skill rather than on the incorporated electrode. In addition, some issues need to address to clarify the results and support publication in Sensors. The reviewer’s comments on the present study are as follows.  

  • Please provide schematics or pictures of the testing systems (with ECG recorders) to enhance readers’ understanding.

Figure was added to Section 2, it is now Figure 5, showing the ECG recording positions on the horse and ECG recording device in the pocket of the textile belt.

  • The authors explained that the aim of the research is to enhance the impedance of textile electrodes through the implementation of specific interconnect systems. However, the results and corresponding discussion are insufficient to clearly address this issue. For example, in Figure 6, the durability of the ECG recorders appears to vary depending on the initial impedance values of the textile electrodes. What is the possible reason for the different start impedance values of each electrode? Please provide the specific solutions to enhance the electrical properties (or impedance) of the textile electrodes in such cases.
    The variation in initial impedance values among textile electrodes can be attributed to differences in factors like electrode-skin contact quality, coat thickness, and skin moisture level. Enhancements to the electrical properties, particularly impedance, can be achieved by optimizing the interconnect systems, such as using conductive gels, modifying electrode size and shape for better conformity, or employing alternative fixation techniques to maintain consistent contact pressure.
  • The authors need to provide the electrical properties of the conductive epoxy. As mentioned by authors, the electrical conductivity is the most significant factor in ensuring reliable electronic signals. This indicates that the intrinsic electrical conductivities of all components and the contact resistance between them are also crucial determinants of the results.

The word epoxy was used incorrectly by mistake and the correct material was added to the Table 2 (nr 7 - Low melting point solder paste - Interflux® DP 5600).

The contact resistance between the paste and the yarn could not be measured as it was integrated directly between laminated fabric layers where curing happened in one step of the lamination. The process is explained now just before Table 2.

  • Long-term ECG recording tests were conducted for 7h 43min, 15h 54min, and 24h until signals were no longer detected. The authors suggested that this possibly due to an increase in impedance at the electrode/fur interface. What were the differences in testing conditions for the electrodes with varying ECG recording durations?
    Differences in testing conditions for long-term ECG recordings can lead to variability in durations primarily due to factors like the thickness of the horse's coat, skin and variations in ambient humidity, affecting skin and electrode moisture levels. Long-term performance could be improved by ensuring adequate hydration either by using moisturizing agents or redesigning electrode features to enhance moisture retention, especially under varying coat and environmental conditions. This was added to the discussion: “Future work could also consider environmental factors like humidity and temperature, which may further impact electrode function over time. To mitigate these, we propose additional testing with moisture-retaining materials with the aim to maintain consistent electrode performance under diverse environmental conditions and coat types.”
  • What is the typical standard for excellent mechanical durability? There are no references for comparison with the results.

Thank you for the comment, this was misleading indeed. The sentence (in section 4 first paragraph) was changed: “The interconnect system shows acceptable durability to washing process (samples could be washed over 20 times) with room for improvement in the sample set stability.”

  • Specific data demonstrating the durability of textile electrodes during long-term ECG recording, in any form, may be required.

We are not sure what data is asked. We specified in the text that electrodes themselves did not change in basic features (resistance from snap fastener to conductive layer) after ECG tests.

  • If the formed electrode system can be applied to human skin, the data trends may differ somewhat from those obtained from horses with fur. Is the goal of the present electrode system intended solely for use with animal? If so, it is necessary to clearly specify the requirements for such measurement conditions.

It was specified in Section 4 and 5.

  • Please check whether the unit ’degrees’ at line 122, should be ‘degree Celsius’.

Changed.

Round 2

Reviewer 2 Report

Comments and Suggestions for Authors

The authors have taken into account all the reviewer's comments. Thanks.